# Production of Porous Biochar from Cow Dung Using Microwave Process

**DOI:** 10.3390/ma16247667

**Published:** 2023-12-15

**Authors:** Wen-Tien Tsai, Li-An Kuo, Chi-Hung Tsai, Hsiang-Lan Huang, Ru-Yuan Yang, Jen-Hsiung Tsai

**Affiliations:** 1Graduate Institute of Bioresources, National Pingtung University of Science and Technology, Pingtung 912, Taiwan; sl.huang98@gmail.com; 2Department of Environmental Science and Engineering, National Pingtung University of Science and Technology, Pingtung 912, Taiwan; sanck112204@gmail.com (L.-A.K.); tsaijh@mail.npust.edu.tw (J.-H.T.); 3Department of Resources Engineering, National Cheng Kung University, Tainan 701, Taiwan; ap29fp@gmail.com; 4Department of Materials Engineering, National Pingtung University of Science and Technology, Pingtung 912, Taiwan; ryyang@mail.npust.edu.tw

**Keywords:** cow dung, microwave pyrolysis, biochar, calorific value, pore property, chemical characteristics, surface functional group

## Abstract

To valorize livestock manure, the present study investigated the production of biochar from cow dung (CD) by microwave pyrolysis. The pore properties and chemical characteristics of CD and CD-based biochar products were found to correlate with the process parameters like microwave power (300–1000 W) and residence time (5–20 min). The findings indicated that CD is an excellent biomass based on the richness of lignocellulosic constituents from the results of proximate analysis and thermogravimetric analysis (TGA). Higher calorific values were obtained at mild microwave conditions, giving the maximal enhancement factor 139% in comparison with the calorific value of CD (18.97 MJ/kg). Also, it can be concluded that the biochar product obtained at 800 W for a holding time of 5 min had the maximal BET surface area of 127 m^2^/g and total pore volume of 0.104 cm^3^/g, which were microporous and mesoporous in the nitrogen adsorption–desorption adsorption analysis. On the other hand, the CD-based biochar contained oxygen-containing functional groups and inorganic minerals based on the spectroscopic analyses by Fourier-transform infrared spectroscopy (FTIR) and energy-dispersive X-ray spectroscopy (EDS), thus featuring to be prone to hydrophilicity in aqueous solutions.

## 1. Introduction

Cow dung, also called cow manure, is mostly composed of digested grass and other smaller amounts of organic/inorganic matters used to feed the ruminants in the cattle’s forage [1]. In Taiwan, over one million metric tons of cow dung (wet basis) were generated annually according to the average values of the daily generation rate (about 25 kg/head) and total cow (dairy cattle) heads on farms (approximately 140,000 heads) [2]. Due to its excellent composition of lignocellulose (or organic matter) and nutrients (e.g., nitrogen), cow dung has been reused as an organic fertilizer, soil amendment, energy source for biogas production, and even packing material in the housing system for dairy cows [3,4,5]. However, these traditional utilization approaches could account for significant emissions of unpleasant smells and greenhouse gases like methane (CH_4_) and nitrous gas (N_2_O) [6,7], posing environmental and health hazards. To achieve higher environmental and economic benefits, advanced manure treatment technologies have to be implemented. Among these advanced technologies, the thermochemical process for conversion of lignocellulosic biomass may be available to efficiently produce biofuels and materials in well-designed refineries. In this regard, the pyrolytic conversion of biomass into biochar materials and its applications have recently been reviewed in the literature [8,9,10,11]. 

Traditionally, pyrolysis refers to the thermal decomposition that occurs in organic material when it is heated by electricity input or fuel combustion in the absence of oxygen (i.e., an inert atmosphere). The process produces solid (charcoal or biochar), liquid (condensable bio-oil), and gaseous products, depending on feedstock type, heating rate, pyrolysis temperature, and its residence time. However, it is generally energy-consuming for economically feasible biorefinery. In recent years, microwave technology has drawn attention for enhancing thermochemical reactions of lignocellulosic residues [12,13,14,15], thus reducing the reaction time and also increasing product yield because heat is generated inside biomass particles by molecular motions [16,17]. It is noteworthy that only a few studies have been conducted using microwave-driven pyrolysis for producing biochar from cow dung [18,19,20]. Zhang et al. investigated the production of biochars from dried cow manure at the target temperatures (300, 400, 500, 600, and 700 °C) for 1 h and also determined the effects of pyrolysis temperature on the total content, chemical speciation, and leaching ability of heavy metals (Cd, Cr, Cu, Ni, Pb, and Zn) [18]. The results showed that the potential risks and ecotoxicity of the resulting biochar were reduced as pyrolysis temperature increased. Luo et al. produced pyrolyzed products (including bio-gas, bio-oil, and biochar) from cow manure using batch and continuous microwave systems with a microwave frequency of 2.45 GHz at a target temperature range of 350–650 °C [19]. Regarding the biochar products, the authors focused on the variations in their thermochemical properties (i.e., elemental analysis and calorific value) with increasing pyrolysis temperature. Nzediegwu et al. studied the fuel, thermal, and surface properties of resulting biochars produced from three lignocellulosic (canola straw, sawdust, wheat straw) and one non-lignocellulosic feedstock (cattle manure pellet) pyrolyzed at three temperatures (300, 400, and 500 °C) using a microwave oven with a frequency of 2.45 GHz [20]. Concerning the pore properties of manure-derived biochars, it was shown that the resulting biochar products had a low specific surface area (3.4–9.7 m^2^/g) based on the Brunauer–Emmett–Teller (BET) theory.

In previous studies [21,22], biochar products from dairy manure were produced by a conventional electricity-resistance heating system at high pyrolysis temperatures (500–900 °C) and holding times of 30 min to enhance its pore and adsorption properties by the removal of cationic compounds (i.e., methylene blue) from water. Furthermore, the production of activated carbon from cow manure by using a potassium hydroxide (KOH) activation process was previously performed in another study [23]. To produce porous biochar products from cow dung (CD) using a microwave pyrolysis process, experiments were performed in a modified microwave oven as a function of output power of 300–1000 W and holding times of 5–20 min in this work. The calorific values, pore properties, surface elemental compositions, and surface functional groups of the resulting CD-derived biochar products were analyzed by adiabatic calorimetry, nitrogen adsorption–desorption isotherms, energy-dispersive X-ray spectroscopy (EDS), and Fourier-transform infrared spectroscopy (FTIR), respectively. In addition, these analytical results were correlated with the process conditions in the microwave pyrolysis system.

## 2. Materials and Methods

### 2.1. Material (i.e., CD)

The starting biomass (i.e., cow dung, herein called CD) for producing porous biochar was collected from the dairy farm of the Taiwan Livestock Research Institute (Hsinhua district, Tainan City, Taiwan). Due to its high moisture (about 80 wt%) and the proneness to stink, this biomass was immediately put into an air-circulating oven at 105 °C to remove the free (or external) moisture. The dried CD sample was further used to determine its thermochemical characteristics, including proximate analysis by the American Society for Testing and Materials (ASTM) test methods, calorific value by adiabatic calorimetry, organic/inorganic elemental contents by energy-dispersive X-ray spectroscopy (EDS), inductively coupled plasma–optical emission spectrometry (ICP–OES), and thermogravimetric analysis (TGA).

### 2.2. Thermochemical Characteristics Analysis of CD

As mentioned above, the thermochemical characteristics of the CD sample were analyzed to determine its relevance to be used as an available biomass for producing carbon materials. In this work, the adopted methods and analytical instruments may refer to previous studies [24,25]. The measurement of inorganic elemental contents was performed using the Agilent 725 ICP–OES instrument (Agilent Co., Santa Clara, CA, USA). Prior to the ICP–OES analysis, a concentrated acid solution (i.e., aqua regia solution) was used to dissolve the dried CD sample into the solution by using a microwave-assisted digestion system. The target elements included aluminum (Al), calcium (Ca), iron (Fe), magnesium (Mg), phosphorus (P), potassium (K), silicon (Si), sodium (Na), sulfur (S), and titanium (Ti). 

### 2.3. Pyrolysis Experiments by Microwave

The pyrolysis experiments by microwave were similar to a previous study [25]. A modified microwave oven (Mei Lin Energy Technology Co., Ltd., Kaohsiung, Taiwan) with an oscillation frequency of 2450 MHz and a maximal power consumption of 1350 W was adopted to perform a series of biochar production experiments under the microwave conditions of output power (300–1000 W) and holding time (5–20 min). In each experiment, approximately 5 g CD was weighed to put into a tube reactor. To ensure an oxygen-free atmosphere, nitrogen gas (500 cm^3^/min) went through the reactor for 5 min prior to the microwave power starting. When finishing each pyrolysis experiment, the resulting biochar was poured out and weighed for the calculation of mass yield. In this work, the resulting biochar products were coded as the notation of BC-CD-power-time. In this regard, BC-CD-600W-10M refers to the CD-based biochar product obtained by applying a power output of 600W and holding for 10 min in the microwave pyrolysis system.

### 2.4. Characterization Analysis of Resulting Biochar

The characterization analysis of the resulting biochar products was analyzed by the following methods: nitrogen adsorption–desorption isotherms for the pore properties (i.e., surface area, pore volume, and pore size distribution), scanning electron microscopy (SEM) for the textural structure on the biochar surface, Fourier-transform infrared spectroscopy (FTIR) for the surface oxygen-containing groups, and energy-dispersive X-ray spectroscopy (EDS) for the surface elemental compositions. The analytical procedures and conditions are described in a previous study [25]. However, the 2D-NLDFT-HS model was used in this work to depict the micropore (pores with diameters less than about 2 nm) size distribution, which was based on the slit–cylinder pore shape boundary fixed at 2 nm [26,27]. 

## 3. Results and Discussion

### 3.1. Thermochemical Properties of DC

Table 1 lists the data on the proximate analysis of the dried DC biomass, showing a high content of volatile matter (81.47 ± 1.62 wt%) and moderate contents of ash (5.97 ± 0.34) and fixed carbon (12.56 wt%). Herein, fixed carbon represents the solid carbon remaining in the DC biomass and the resulting biochar without the loss (or devolatilization) during the pyrolysis process. As compared to the data on fixed carbon for various types of biomass, this CD biomass had a lower amount [28,29]. Based on the typical calorific values of representative biomass and coal [28], the calorific value (18.97 ± 0.41 MJ/kg) showed a reasonable value because the CD contained herbaceous biomass. The higher the ash value, the lower the organic matter. In this regard, the calorific value of the CD biomass was higher than that of rice residues (rice straw or rice husk) and cow manure [18,19,20] but was lower than those of woody biomass residues. Energy-dispersive X-ray spectroscopy (EDS) analysis also showed that the main elemental contents of the CD feedstock included carbon (C, 47.58 wt%), oxygen (O, 49.77 wt%), and calcium (Ca, 2.46 wt%), which could be derived from lignocellulosic biomass and calcium-containing minerals (e.g., calcite or calcium oxide) [28,29]. The contents of inorganic elements in the CD biomass are listed in Table 2. The results showed that many metal oxides were present in the CD ash, but their compositions were quite diverse. As expected, the oxides of calcium (Ca), silicon (Si), and potassium (K) were dominant, which were consistent with those of herbaceous species [29,30]. It should be noted that a significant sulfur content was found in the CD biomass, which could cause the formation of corrosive gases (e.g., H_2_S, SO_x_) and slagging and fouling in the thermochemical process. Figure 1 further depicts the thermal decomposition behavior of the CD biomass by the curves of thermogravimetric analysis (TGA) and derivative thermogravimetry (DTG) at heating rates of 5, 10, 15, and 20 °C/min under a nitrogen flow (i.e., 50 cm^3^/min). These curves were very similar to those of lignocellulosic species, showing two devolatilization peaks at about 300–350 °C and 350–400 °C, depending on the heating rates and lignocellulosic compositions [31]. In general, the first devolatilization should be due to the thermal decomposition of the hemicellulose content because it tends to yield more gases and less tar than cellulose and lignin at lower pyrolysis temperatures. In this work, the microwave pyrolysis system was operated above 450 °C to produce fully carbonized and charred products.

### 3.2. Mass Yield and Calorific Value of CD-Based Biochar Products

The mass yield of CD-based biochar was weighed by the ratio of its residual mass to CD mass fed into the microwave pyrolysis system. According to the experimental design, there were thirteen CD-based biochar products, which were obtained by a microwave output power of 300–1000 W and residence time of 5–20 min in this work. Due to more pyrolysis reactions at larger output powers and longer residence times, the mass yields ranged from 15 wt% to 30 wt% and also showed a declining trend as a result of the evolution of volatiles and other degradable components from CD. For example, the order of the mass yield is given below: 29.85 wt% (BC-DC-300W-10M), 24.37 wt% (BC-DC-440W-5M), 22.03 wt% (BC-DC-440W-10M), and 15.81 wt% (BC-DC-800W-5M). Obviously, the mass yields of CD-based biochar products were lower than those of biochar products from rice husk in a previous study [25], which could be attributed to the different ash contents between CD and rice husk. On the other hand, the calorific values of CD-based biochar products were also determined to show their enhancement factors. The findings showed that higher calorific values were obtained under mild microwave conditions, which were consistent with those in a previous study [25]. For example, the calorific values of 25.96 MJ/kg and 26.32 MJ/kg were given for the biochar products of BC-CD-300W-10M and BC-CD-440W-5M, respectively. Correspondingly, the enhancement factors were 137% and 139% in comparison with the calorific value of CD (18.97 MJ/kg, listed in Table 1). This result can be related to the progressive reduction in the oxygen content of CD-based biochar product as microwave power was applied. However, the microwave pyrolysis conditions at larger power outputs and/or longer residence times were unfavorable for producing biochar products with higher calorific values.

### 3.3. Pore Properties of CD-Based Biochar Products

Table 3 lists the BET surface area, total pore volume, micropore surface area, and micropore volume of CD-based biochar products. These values were obtained by their N_2_ adsorption–desorption isotherms at −196 °C (Figure 2). To feature the characteristics of microporous and mesoporous structures, Figure 3 and Figure 4 further depict their pore size distributions based on the Barrett–Joyner–Halenda (BJH) method [32,33] and the 2D-NLDFT-HS model [26,27], respectively. Based on the results in Table 3 and Figure 3 and Figure 4, the key findings are summarized and discussed below: The pore properties of CD-based biochar products indicated an increasing trend as the microwave output power increased from 300 to 800 W at a holding time of 5 min, suggesting an increase in the surface area and pore volume due to more pore development. However, the pore properties may be reduced because of a severe pyrolysis reaction at larger output powers and/or longer residence times. In this work, the CD-based biochar product (i.e., BC-CD-800W-5M) had the maximal BET surface area of 126.99 m^2^/g and total pore volume of 0.104 cm^3^/g, which were obtained at a microwave output power of 800 W and a holding time of 5 min.According to the International Union of Pure and Applied Chemistry (IUPAC) classification of physical adsorption isotherms [32,33], the CD-based biochar products are characteristic of Type I and Type VI in Figure 2, which are indicative of microporous and mesoporous structures, respectively. The former refers to the adsorption of nitrogen (adsorbate gas) molecules to the adsorbent with micropores, which are covered with a monolayer of adsorbed molecules on the surface of the adsorbent at a very low relative pressure. By contrast, the hysteresis loops (Type VI isotherms) were seen to start from approximately 0.45 relative pressure in the nitrogen desorption isotherms. According to the IUPAC classification of hysteresis loops, they should belong to Type H4 loops [32,33], which are indicative of mesoporous solids with narrow slits. Therefore, the BJH method was used to calculate their mesopore size distributions (seen in Figure 3) based on the N_2_ desorption isotherm data. The peak at about 3.8 nm featured the mesopores (pores with a diameter or width of 2–50 nm) in the CD-based biochar products.Based on the pore properties in Table 3 and the adsorption–desorption isotherms in Figure 2, the microscale structures of the resulting biochar products were mainly microporous. In this regard, the 2D-NLDFT-HS model was adopted to depict the micropore size distribution of the optimal biochar product (i.e., BC-CD-800W-5M) with slit-shaped pores [34], as shown in Figure 4. It can be seen from this model’s analysis that there are two peaks at about 0.70 nm and 0.95 nm, which are lower than the micropore boundary (i.e., 2.0 nm).

### 3.4. Chemical Characteristics of CD-Based Biochar

To obtain information about the variations in chemical characteristics of CD-based biochar, spectroscopic analytical tools, including Fourier-transform infrared spectroscopy (FTIR) and dispersive X-ray spectroscopy (EDS), were used in this work. Figure 5 depicts the FTIR spectrum of the optimal biochar (i.e., BC-DC-800W-5M) in comparison with its starting biomass (i.e., CD) diluted in KBr to give more resolved spectral features. It clearly showed that there were significant differences between the biochar and CD in the patterns of oxygen-containing functional groups. Due to the highly increased aromatic structure of the biochar sample produced at high temperature, the spectrum of the resulting biochar seemed to become more refractory, with no strong peaks. The most dramatic change was the hydroxyl (O-H) stretch peak at around 3433 cm^−1^, which almost disappeared in the biochar spectrum. It implied that significant oxygen-containing functional groups are formed in the biochar by microwave pyrolysis. Using the data on the functional groups of carbon materials [35,36,37,38], these peaks can be associated with the corresponding oxygen/carbon-containing functional groups. The important peaks in the biochar spectrum could be alkyne C≡C (2283 cm^−1^), C=C stretching (1639 cm^−1^), O–H bending (1384 cm^−1^), and C-O stretching (1110 cm^−1^).

In this work, spectroscopic analysis by energy-dispersive X-ray spectroscopy (EDS) was used to examine the surface elemental compositions of the resulting biochar semi-quantitatively. Figure 6 shows the EDS spectrum of the optimal biochar (i.e., BC-DC-800W-5M), revealing an increased amount of carbon (C, 83.1 wt%) and a reduced content of oxygen (O, 9.2 wt%) on the sample surface as compared to those of the starting CD biomass (C, 47.5 wt%; O, 49.8 wt%). These results could be attributed to the devolatilization of non-carbon elements (i.e., oxygen and hydrogen) from the starting lignocellulose during the microwave pyrolysis process as volatiles or gases, resulting in the reduction in oxygen content and the increase in calorific value. On the other hand, the resulting biochar also contained smaller amounts of inorganic elements (including calcium, silicon, potassium, and magnesium), which were present in ash in the forms of metal oxides like CaO, SiO_2_, K_2_O, and MgO. Although data from EDS can be analyzed in a short time, great care must be taken to ensure they are reliable and reproducible. From the analytical results by FTIR and EDS, these oxygen-containing functional groups and inorganic minerals could increase the hydrophilicity and cation exchange capacity (CEC) of CD-based biochar. Due to the features of high porosity and hydrophilicity of the resulting biochar, it is advantageous for the removal of cationic compounds and metal ions from aqueous solutions as an excellent adsorbent. 

## 4. Conclusions

The present work investigated the production of porous biochar from cow dung (CD) by microwave pyrolysis to valorize livestock manure in an efficient approach. The highest calorific values were obtained at mild microwave conditions, giving a maximal enhancement factor of 139% in comparison with the calorific value of CD (18.97 MJ/kg). Therefore, CD-based biochar can be used as an auxiliary coal-like fuel in industrial boilers. In addition, the findings showed that the biochar product obtained at 800 W at a holding time of 5 min had the maximal BET surface area of 127 m^2^/g and total pore volume of 0.104 cm^3^/g, which indicated a porous carbon material with microporous and mesoporous structures. Due to the presence of oxygen-containing functional groups and inorganic minerals, the porous CD-based biochar indicated hydrophilicity on the surface, suggesting potential applications in the removal of cationic compounds or metal ions from aqueous solutions.

## Figures and Tables

**Figure 1 materials-16-07667-f001:**
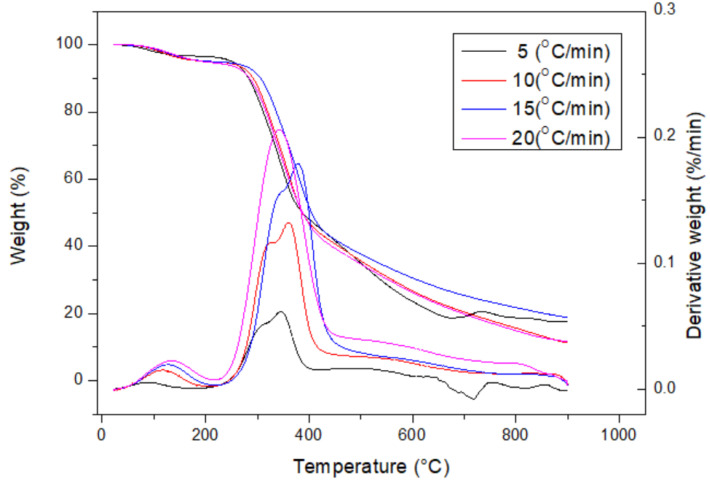
Thermogravimetric analysis (TGA) and derivative thermogravimetry (DTG) curves of dried CD sample at heating rates of 5, 10, 15, and 20 °C/min.

**Figure 2 materials-16-07667-f002:**
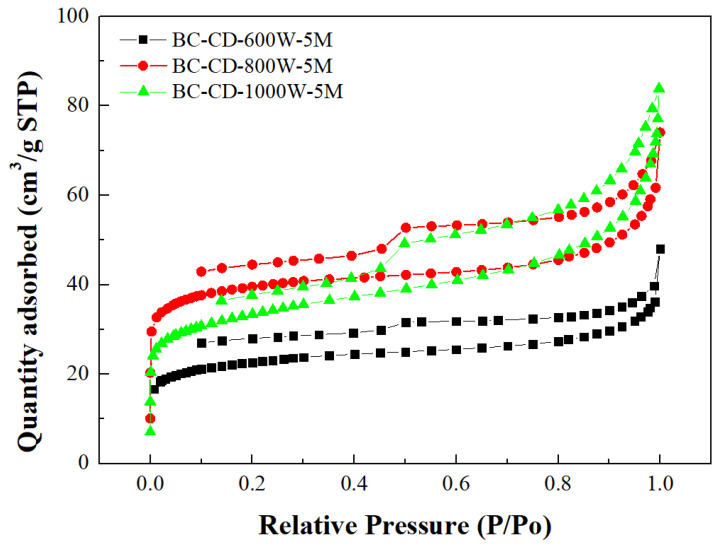
N_2_ adsorption–desorption isotherms of resulting biochars produced from cow dung (i.e., CD) at microwave powers of 600–1000 W and fixed residence time of 5 min.

**Figure 3 materials-16-07667-f003:**
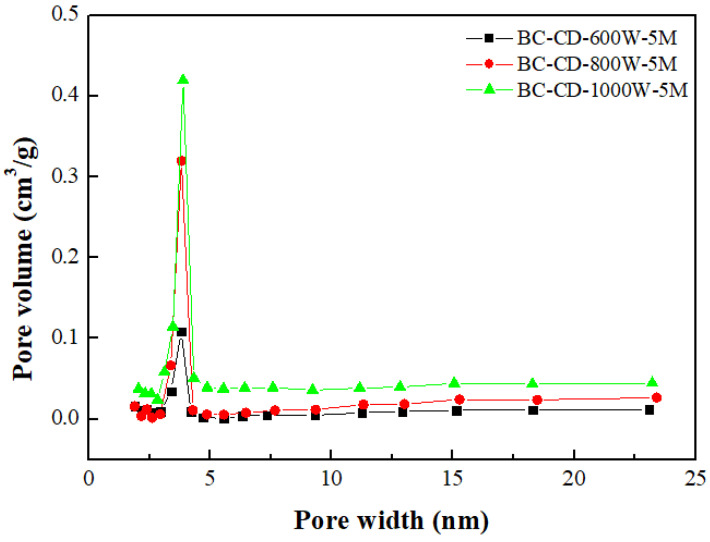
Mesopore size distributions of resulting biochars produced from cow dung (i.e., CD) at microwave powers of 600–1000 W and fixed residence time of 5 min.

**Figure 4 materials-16-07667-f004:**
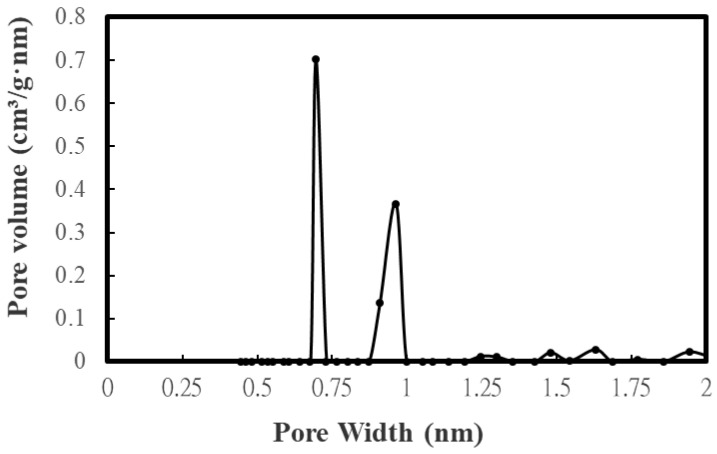
Micropore size distributions of the optimal biochar (i.e., BC-DC-800W-5M).

**Figure 5 materials-16-07667-f005:**
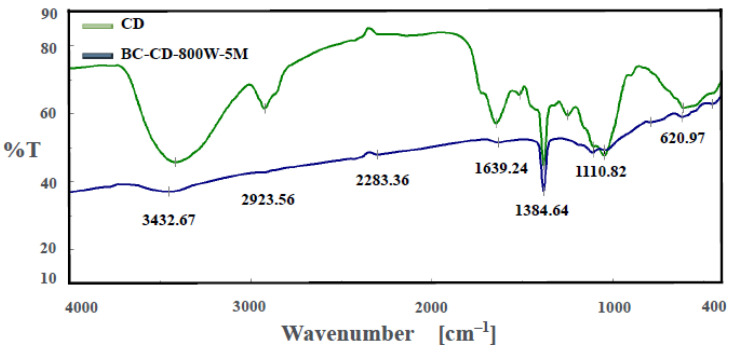
Fourier-transform infrared spectroscopy (FTIR) spectra of CD and biochar (i.e., BC-DC-800W-5M).

**Figure 6 materials-16-07667-f006:**
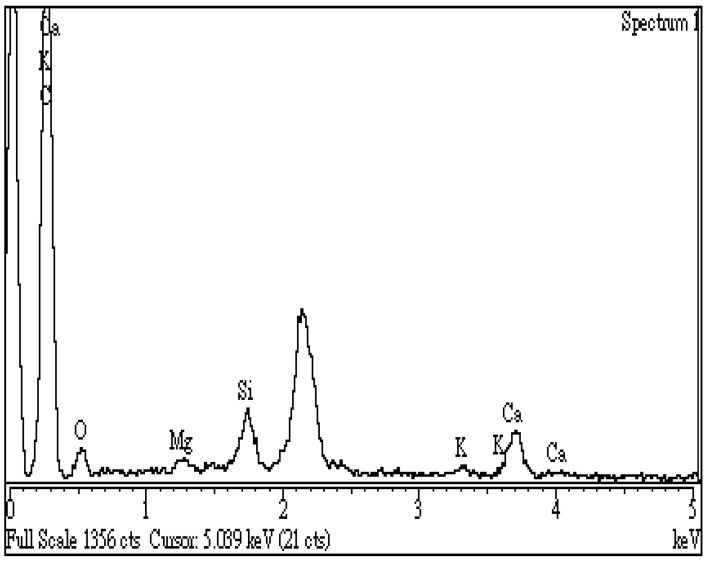
Energy-dispersive X-ray spectroscopy (EDS) spectrum of the optimal biochar (i.e., BC-DC-800W-5M).

**Table 1 materials-16-07667-t001:** Proximate analysis and calorific value of cow dung (CD).

Property	Value
Proximate analysis ^a,b^	
Ash (wt%)	5.97 ± 0.34
Volatile matter (wt%)	81.47 ± 1.62
Fixed carbon ^c^ (wt%)	12.56
Calorific value (MJ/kg) ^a,b^	18.97 ± 0.41

^a^ Mean ± standard deviation for three determinations; ^b^ the values were based on the dry basis; ^c^ by difference.

**Table 2 materials-16-07667-t002:** Inorganic elemental contents of cow dung (CD).

Inorganic Element	Value	Method Detection Limit (ppm)
Sulfur (S)	1.556 wt%	14.4
Calcium (Ca)	1.309 wt%	1.0
Silicon (Si)	1.109 wt%	4.7
Potassium (K)	0.882 wt%	0.2
Phosphorus (P)	5204 ppm	1.0
Aluminum (Al)	1640 ppm	0.1
Sodium (Na)	964 ppm	0.2
Iron (Fe)	827 ppm	0.1
Magnesium (Mg)	233 ppm	0.1
Titanium (Ti)	120 ppm	0.1

**Table 3 materials-16-07667-t003:** Pore properties of CD-derived biochar products.

CD-Derived Biochar ^a^	S_BET_ ^b^ (m^2^/g)	S_micro_ ^c^ (m^2^/g)	V_t_ ^d^ (cm^3^/g)	V_micro_ ^c^ (cm^3^/g)
BC-CD-300W-5M	50.58	30.42	0.045	0.016
BC-CD-300W-10M	18.13	15.50	-- ^e^	0.008
BC-CD-300W-20M	2.22	8.41	0.001	-- ^e^
BC-CD-440W-5M	71.51	43.55	0.067	0.022
BC-CD-440W-10M	113.81	76.08	0.082	0.040
BC-CD-440W-20M	99.86	56.99	0.081	0.030
BC-CD-600W-5M	74.10	41.90	0.065	0.021
BC-CD-600W-10M	95.50	58.95	0.081	0.031
BC-CD-600W-20M	86.87	57.24	0.072	0.030
BC-CD-800W-5M	126.99	87.10	0.104	0.044
BC-CD-800W-10M	91.44	51.13	0.087	0.026
BC-CD-1000W-5M	108.61	50.70	0.119	0.263
BC-CD-1000W-10M	57.31	42.49	0.059	0.027

^a^ Product notation indicated the resulting biochar pyrolyzed at a microwave power of 300–1000 W and holding time of 5–20 min using 5 g CD. ^b^ BET surface area (S_BET_) was based on relative pressure range of 0.05–0.30. ^c^ Total pore volume (V_t_) was obtained at relative pressure of about 0.995. ^d^ Micropore surface area (S_micro_) and micropore volume (V_micro_) were obtained by using the *t*-plot method. ^e^ Not available because of the limited pore properties.

## Data Availability

Data are contained within the article.

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
