# Peer review of "Production of Porous Biochar from Cow Dung Using Microwave Process"

_materials, 2023, doi:10.3390/ma16247667_

Round 1

Reviewer 1 Report

Comments and Suggestions for Authors

1.       Please explain how samples were mineralized for ICP-OES analysis.

2.       How was the elemental composition determined in Table 2?

3.       Were stable temperature increases achieved in the thermogravimetric system? If not, what deviations occurred?

Author Response

Q1. Please explain how samples were mineralized for ICP-OES analysis.

Reply: In this work, the description about the mineralization of the CD sample has been added to the revised manuscript.  In addition, the authors also appreciated the National Tsing Hua University for the analysis in the Acknowledgments.

“Prior to the ICP-OES analysis, the concentrated acid solution (i.e., aqua regia solution) was used to dissolve the dried CD sample into the solution by using a microwave-assisted digestion system.

Q2. How was the elemental composition determined in Table 2?

Reply: Regarding the elemental compositions in Table 2 determined by the energy dispersive X-ray spectroscopy (EDS), it is based on the X-rays emitted from the sample, measuring their energy and thus generating a spectrum of X-ray energy versus intensity (not shown in the manuscript).  The observation of peaks in the spectrum can be performed through the system software in the EDS to quantify the elemental compositions as weight percentages.  Although the data from the EDS can be performed in a short time, great cares must be taken to ensure them reliable and reproducible.

Q3. Were stable temperature increases achieved in the thermogravimetric system? If not, what deviations occurred?

Reply: Regarding the thermogravimetric analysis (TGA) system, it is a useful tool for measuring the relationship between the sample mass and its temperature (or the variations on test material mass with hanging temperature) as the sample was heated in an inert atmosphere.  To find the thermal decomposition features, mass loss thermograms were collected and normalized based on the initial sample mass using the TGA instrument software, thus giving TGA curves (% mass loss versus temperature or time).  Furthermore, the first derivative of the thermograms were calculated to produce the derivative thermogravimetry (DTG) curves (seen in Figure 1).

Reviewer 2 Report

Comments and Suggestions for Authors

The manuscript "Production of porous biochar from cow dung using microwave process "is very well written and designed although I can suggest some improvements to be done before eventual publication.

Figure legends are very short and little informative with many abbreviations which is not good for respective readers. Please make them informative and descriptive.

Figures are not uniform, all have different styles, different legend fonts, legend sizes etc. I understand many of them are exported from the device they were obtained from but please reconsider making them uniform according to journals requirements.

The chapter "3.3. Chemical characteristics of CD-based biochar" that presents FTIR and X-ray spectroscopy data lack serious discussion. Please provide in depth discussion.

Author Response

Q1. Figure legends are very short and little informative with many abbreviations which is not good for respective readers. Please make them informative and descriptive.

Reply: As suggested by the reviewer, the legend in Figure 1 has been revised to make them informative and descriptive.

Q2. Figures are not uniform, all have different styles, different legend fonts, legend sizes etc. I understand many of them are exported from the device they were obtained from but please reconsider making them uniform according to journals requirements.

Reply: As suggested by the reviewer, Figures have been revised to make it clear and consistent.

Q3. The chapter "3.3. Chemical characteristics of CD-based biochar" that presents FTIR and X-ray spectroscopy data lack serious discussion. Please provide in depth discussion by marking red color.

Reply: As suggested by the reviewer, the discussions on about the FTIR and X-ray spectroscopy data have been additionally addressed to provide more significant results.

“To obtain the information about the variations on chemical characteristics of CD-based biochar, the spectroscopic analytical tools, including Fourier Transform infrared spectroscopy (FTIR) and dispersive X-ray spectroscopy (EDS), were used in this work.  Figure 5 depicted the FTIR spectrum of the optimal biochar (i.e., BC-DC-800W-5M) in comparison with its starting biomass (i.e., CD) diluted in KBr for giving more resolved spectral features.  It obviously showed that there were significant differences between the biochar and CD in the patterns of oxygen-containing functional groups.  Due to its highly increased aromatic structure for the biochar sample produced at high temperature, the spectrum of the resulting biochar seemed to become more refractory, not have strong peaks.  The most dramatic change was the hydroxyl (O-H) stretch peak at around 3433 cm-1, which was almost disappeared in the biochar spectrum.  It implied that the significant oxygen-containing functional groups will be formed in the biochar by microwave pyrolysis.  Using the data on the functional groups of carbon materials [35-38], these peaks can be associated with the corresponding oxygen/carbon-containing functional groups.  The important peaks in the biochar spectrum could be alkyne C≡C (2283 cm−1), C=C stretching (1639 cm−1), O–H bending (1384 cm−1), and C-O stretching (1110 cm−1).

In this work, the spectroscopic analysis by the energy dispersive X-ray spectroscopy (EDS) was used to examine the surface elemental compositions of the resulting biochar semi-quantitatively.  Figure 6 showed the EDS spectrum of the optimal biochar (i.e., BC-DC-800W-5M), revealing increased amount of carbon (C, 83.1 wt%) and reduced content of oxygen (O, 9.2 wt%) on the sample surface as compared to those of the starting biomass CD (C, 47.5 wt%; O, 49.8 wt%).  These results could be attributed to the devolatilization of non-carbon elements (i.e., oxygen and hydrogen) from its starting lignocellulose during the microwave pyrolysis process as volatiles or gases, resulting in the reduction in oxygen content and the increase in calorific value.  On the other hand, the resulting biochar also contained less amounts of inorganic elements (including calcium, silicon, potassium, and magnesium,), which were present in ash as the forms of metal oxides like CaO, SiO2, K2O, and MgO.  Although the data from the EDS can be performed in a short time, great cares must be taken to ensure them reliable and reproducible.  From the analytical results by FTIR and EDS, these oxygen-containing functional groups and inorganic minerals could increase the hydrophilicity and cation exchange capacity (CEC) of CD-based biochar.  Due to the features for the high porosity and hydrophilicity of resulting biochar, it is advantageous for removal of cationic compounds and metal ions from aqueous solution as an excellent adsorbent.”
